# RNA Editing in Chloroplast: Advancements and Opportunities

Taimyiah Mohammed [1,*], Ahmad Firoz [1,2] and Ahmed M. Ramadan [1,2,3,*]

1   Department of Biological Sciences, Faculty of Science, King Abdulaziz University (KAU),
    P.O. Box 80141, Jeddah 21589, Saudi Arabia
2   Princess Dr. Najla Bint Saud Al-Saud Center for Excellence Research in Biotechnology,
    King Abdulaziz University, Jeddah 21589, Saudi Arabia
3   Agricultural Genetic Engineering Research Institute (AGERI), Agriculture Research Center (ARC),
    Giza 12619, Egypt
*   Correspondence: tsalehmohammed@stu.kau.edu.sa (T.M.); aamara@kau.edu.sa (A.M.R.)

**Abstract:** Many eukaryotic and prokaryotic organisms employ RNA editing (insertion, deletion, or conversion) as a post-transcriptional modification mechanism. RNA editing events are common in these organelles of plants and have gained particular attention due to their role in the development and growth of plants, as well as their ability to cope with abiotic stress. Owing to rapid developments in sequencing technologies and data analysis methods, such editing sites are being accurately predicted, and many factors that influence RNA editing are being discovered. The mechanism and role of the pentatricopeptide repeat protein family of proteins in RNA editing are being uncovered with the growing realization of accessory proteins that might help these proteins. This review will discuss the role and type of RNA editing events in plants with an emphasis on chloroplast RNA editing, involved factors, gaps in knowledge, and future outlooks.

**Keywords:** plant chloroplast; RNA editing; PPR proteins





## 1. Introduction

RNA is transcribed from DNA using RNA polymerase in the nucleus, mitochondria and plastids and is further processed if needed. RNA editing represents a post-transcriptional modification (insertion, deletion or conversion) mechanism observed in many domains of life, including prokaryotes and eukaryotes [1]. The editing of RNA has already been identified in many plants, such as Arabidopsis, rice, soybean, tobacco, wheat, barley, and maize [2–4]. This is a powerful genetic phenomenon in organisms and was first discovered when Benne (1986) was investigating the mitochondrial cytochrome oxidase in the protozoan Trypanosoma brucei [5,6].

To begin with, RNA is transcribed normally as per the genetic code of the gene. After this, the RNA is altered in certain ways, which can incorporate uridine at certain loci and/or facilitate a change of cytidine to uridine [7]. These editing events can happen in most forms of RNA, including messenger RNA (mRNA), transfer RNA (tRNA), and ribosomal RNA (rRNA), transcribed in the cell, including organelles such as mitochondria and chloroplast [8]. The final result after such editing events might ultimately generate an RNA molecule that does not entirely reflect the complimentary genetic code in the DNA from which it was deciphered [9]. Numerous studies have shown that RNA editing events could lead to impaired organelle biogenesis, endosperm development, and response to stress conditions in plants, and thus, it becomes important to study it [10–12].

RNA editing can happen in multiple ways, such as the C-to-U, U-to-C, and A-to-I conversions, as well as insertions and deletions of U or G. However, in plants, RNA editing occurs mainly in the mitochondria and plastids. C-to-U-type RNA editing is primarily found [8]; however, recent studies have investigated new types of editing without suggesting the mechanisms involved in these new types, such as U to G [13]. Plant organellar

RNA editing involves at least three elements: trans-acting factors that bind to cis-acting elements and thereby recruit editing enzymes to induce nucleotide conversion [14]. Plant RNA editing is mediated by a variety of pentatricopeptide repeat (PPR) proteins encoded in the nucleus [15,16]. The editosome machinery of flowering plants also requires several other non-PPR protein factors, such as RNA interacting protein/multiple organellar RNA editing factor (RIP/MORF family), organelle RNA recognition motif (ORRM family), and organelle zinc finger (OZ family) [16]. Advancements in gene sequencing, computational methods, and data analysis pipelines have led to high-resolution access to RNA editing events and predictions towards it [17]. Many online and offline resources are available to predict and assist researchers in searching for RNA editing events in their study organism and comparing them with other plants with an evolutionary background. Phylogenetic analysis, statistical correlations and regression models, machine learning approaches and artificial intelligence based on available datasets further empower the research into RNA editing and understanding it in a larger context [18].

This study gives an overview of transcription and RNA editing in plant chloroplasts, the role of involved proteins, and their mechanism. Additionally, some useful bioinformatics tools are available to study to help the researcher to select a suitable one or a combination of them to study RNA editing in different organisms

## 2. Transcription and RNA Editing in Plant Chloroplasts

The expression of chloroplast DNA (cpDNA) genes depends on post-transcriptional changes such as polycistronic mRNA processing, intron splicing, and RNA editing [19]. Angiosperm chloroplast genes are transcribed by bacterial-like multi-subunit RNA polymerase (PEP) and phage-like RNA polymerase (NEP) [20]. The PEP enzyme is made from subunits derived from nuclear and cpDNA genes. The subunits α, β, β′ and β″ are encoded by cp genes. However, sigma (SIG) factors and polymerase-associated proteins are encoded by nuclear genes [21]. These types of RNA polymerases are recognized at bacterial-like promoters in chloroplast genome-like *psaS* genes [22]. NEP is a single subunit enzyme that has similarity to bacteriophage-type RNA polymerase, which is responsible for promoting the transcription of cp PEP subunits [21]. There are three types of NEP polymerases coded by *RPOT* genes: *RPOTp*, *RPOTm* and *RPOTmp* [20,22] (Figure 1). The NEP polymerases are divided into three class (Ia, Ib and II) depending on the features of recognized promoters [22].

The factors that control the chloroplast gene transcription originate from the nucleus and the chloroplast. The nuclear factors PAPs are essential for the regulation of transcription, whereas SIGs are important for PEP polymerase binding to the promoter [23]. Recent studies have also shed light on the importance of novel signaling molecules called alarmones, which are produced during the stress response of plants [24]. Other factors that affect the chloroplast gene expression include chloroplast RNA editing.

RNA editing is generally perceived as a repair tool to correct genomic mutation (point mutation) at the transcript level in organelles such as mitochondria and chloroplasts [20]. It has been observed that many genes, such as ndhD-2, rpoA, rps12, etc., have altered levels under different environmental conditions [2,25,26]. The presence of different protein isoforms with different characteristics in terms of stability and functionality is observed simultaneously, which dictates the cellular metabolism and stress response. Moreover, RNA editing can happen in non-coding regions, which can influence many vital events in cellular homeostasis within the cell [27]. Intriguingly, the degree of RNA editing varies across different organs, tissues, mutant lines, environmental factors and developmental stages [11]. The mode of RNA editing also seems to vary between various plant species and even among variants of the same species of plants [28].

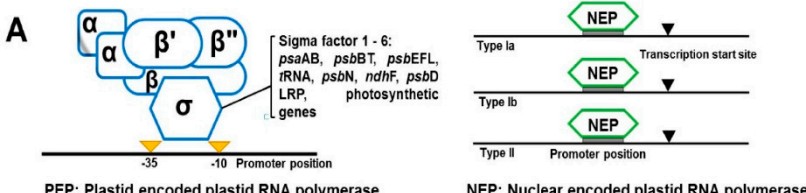

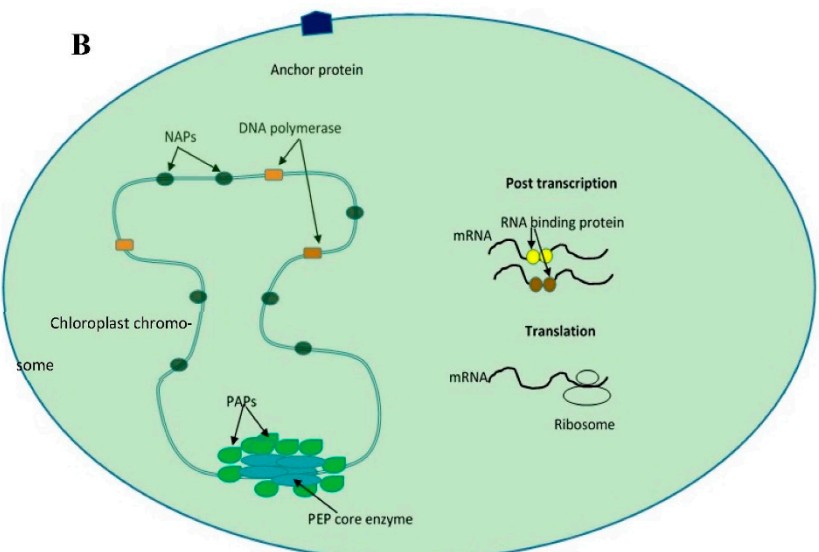

**Figure 1.** The figures show the general scheme of chloroplast transcription. (**A**) shows the two types of chloroplast RNA polymerase transcription that occurs in plants: plastid-encoded plastid RNA polymerase (PEP, which recognizes bacterial-like promoters such as PSA gene promoters) and nucleus-encoded plastid RNA polymerase (NEP), which recognizes housekeeping genes in chloroplast-like *rpo* genes. NEB is divided into subclasses Ia, Ib and II depending on the structure of recognized promoters. (**B**) shows the general scheme of plants' transcription that involves relevant proteins in nucleoid. PEP: plastid RNA polymerase, white blue color; PAPs: RNA polymerase-associated proteins, green color; NAPs: plastid nucleoid-associated proteins; mRNA binding proteins: such as PPR and mTERF, yellow and brown color.

With the advancement of high-throughput next-generation sequencing technologies, many plant organelle (mitochondria and chloroplast) genomes have been sequenced, and they have shown a very high number of RNA editing events, showing the extensiveness of this phenomenon in the plant genome [29]. A surprising result was that more than 3000 C-to-U RNA editing events were identified while analyzing the chloroplast transcriptomic profile of *Selaginella uncinata* [30]. Using RNA-seq technology, about 41 C-to-U altered loci were identified inside the transcripts of the chloroplast genome of *Vigna radiata* [31]. An analysis of the complete plastid transcriptomes of moth orchids led to the identification of 137 editing sites, which showed that 93 out of 137 were novel editing events, and 79 of 137 were present in the protein-coding region [32]. The impact of temperature on the RNA editing events was investigated using deep genome sequencing information in grape organelles. The results were as follows: 627 RNA editing events were recognized in mitochondria, and 122 RNA editing events were recognized in chloroplasts [33]. In addition, a general trend was discovered in which the expression level of many PPR genes and the editing frequency were negatively correlated with temperature. This means that at higher temperatures, RNA editing efficiency is decreased. Additional studies further confirmed that RNA editing events were affected at higher temperatures during natural heat stress in grapes [33]. The

biogenesis of chloroplasts in plants also seems to be affected by RNA editing through the interference of chloroplast development and seedlings' growth [31].

## 3. RNA Editing in Different Types of RNA

After the generation of each type of RNA, it is processed to remove unnecessary sequences at both ends and introns (in mRNA) and make the final mature RNA. However, these RNAs are also subjected to modifications after their maturation, and RNA editing is one such modification [34]. Depending upon the type of RNA (mRNA, tRNA, or rRNA) on which the editing events occur, RNA editing could be classified. Editing in the mRNA is the most common and widely studied type of RNA editing observed across viruses, prokaryotes and eukaryotes [35]. mRNA editing can involve a change in the existing base in the protein-coding sequence (mRNA), uridine insertion and cytidine-to-uridine (C-to-U) conversions to generate new stop codons, or, in many instances, the start/stop codons are introduced/removed to shift the open reading frame and thus the translation process [36]. These changes can result in the formation of new start/stop codons that lead to the creation of novel reading frames and, consequently, altered protein structures [8].

The tRNAs ensure proper duty and folding of the mRNA, and tRNA editing involves several base modifications that have been reported in marsupials, splizellomyces, myxomycetes, Leishmania, metazoans, land snails, squids, bacteria, and land plants [37]. This can lead to altered tRNA identity and the creation of new substrates. The editing can create changes at all levels involving loop nucleotides and acceptor stems and might affect the dynamics of mRNA molecules [38,39]. The rRNA is the most abundant (about 80%) RNA inside the cell, but RNA editing seems to have less frequency than that of protein-coding transcripts, and only a few such sites have been reported in plants, such as in *Selaginella moellendorffii* [40] and *Pulsatilla patens* [41].

RNA editing events could be predicted by comparing the genomic sequences and amino acid sequences for a particular plant. Further validation of editing could be done using quantitative PCR for specific genes. Reports have also shown little (5–12%) RNA editing in the untranslated and intron sequences of the genome [42].

## 4. Pentatricopeptide Repeat Proteins

Focused studies done on flowering plants have identified cis-acting genetic elements (20–25 bases upstream of the edit site) near the RNA edit sites, and some putative site-specific proteins have been discovered that interact with these cis-acting elements [5]. Editing occurs primarily when cytidines are deaminated into uridines (C-to-U), a result of specific nucleus PPR proteins [15].

Apart from this, trans-acting elements were also identified accidentally, such as chlororespiratory reduction 4 (CRR4) protein, which belongs to the pentatricopeptide repeat (PPR) protein family [43]. PPR proteins are very diverse inside plants, but they have similarities to proteins that mediate protein-protein interactions, mRNA processing, and proteins involved in developmentally defective chloroplast function and biogenesis, pigmentation, and the development of embryos [44]. PPR proteins can be broadly classified as P and PLS classes based on the structures of their motifs (Figure 2). The P-class PPR proteins are involved in the stress response of plants [45]. Any defect in PPR protein has been associated with 35 amino-acid-long loosely conserved repeats (PPR or P motifs) that form an active site that identifies the RNA target. In the case of the PLS class, they contain the canonical P motifs, followed by conserved plant-specific domains (E) and the sometimes 100 amino-acid-long DYW domain, at their C-terminus [46].

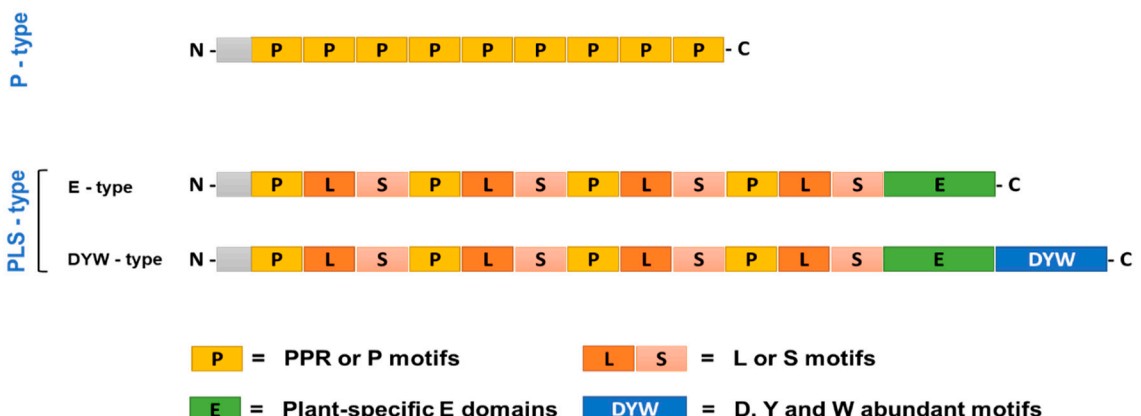

**Figure 2.** The general architecture of three types of pentatricopeptide repeat (PPR) proteins. The P-type has 35-amino-acid repeats that end with proline (this type is involved in RNA stability). The PLS-type (this type is involved in RNA editing) has three types of repeats (P repeats, S repeats, which are shorter than P repeats, and L repeats, which are longer than P repeats) showing P, L, and S motifs. E/DYW domains have deamination activity.

By binding to RNA and preventing its degradation by cellular exonucleases, the P-type PPR proteins are thought to be involved in RNA stabilization. In *Arabidopsis thaliana*, a P-type PPR protein known as BFA2 was found to prevent the degradation of atpH/F transcripts within chloroplasts [47]. P-type PPR proteins are also implicated in assisting the splicing of group II introns of plant organelles, which otherwise cannot undergo self-splicing and need assistance. For example, a P-type PPR protein, THA8, was reported, which binds and assists in the splicing of ycf3 intron 2 in *Brachypodium distachyon* [48]. Another P-type PPR protein, EMB-7L, is involved in the chloroplast genes' RNA splicing and is essential for normal embryo development in maize [49]. The roles of the P-type PPR proteins AtPPR4 and OsPPR4 were found to take part in the splicing of rps12 introns in A. thaliana and O. sativa, respectively, which was essential for plant development and chloroplast biogenesis [50]. Similarly, a novel P-type PPR protein, ECD2, was identified, which was involved in the assisted splicing of many group II introns in *A. thaliana* and was vital during its early development [51].

As demonstrated in *A. thaliana* [52] and rice [53], the additional C-terminal domains (E and DYW) of PLS-type PPR proteins are necessary for RNA editing. It is believed that the plant-specific E domains' PPR-like motifs facilitate interaction with other proteins [54]. The zinc-binding motif (HxE(x)nCxxC) that is also present in numerous cytidine deaminases can be found in abundance in the D, Y, and W amino acids of the DYW [55]. Although deamination activity has not been demonstrated, this DYW motif has been found to occasionally bind the "C" nucleotide. However, studies in *A. thaliana* demonstrated that DYW1 functions as an RNA editing factor and interacts with the ndhD-1 gene [56]. This DYW1 has a partial E domain and a conserved DYW domain. It works with an E-type PPR protein to make sure that ndhD-1 mRNA changes from ACG to AUG. Five additional proteins that are very similar to DYW1 have been found, which suggests that the DYW1 protein may frequently interact with the E-type PPR protein in A. thaliana [57].

## 5. Mechanisms of RNA Editing

Although many clues about the factors involved have been found, we still do not fully understand how RNA editing occurs in plants. Recently, a code for PPR-RNA recognition has been described, where each PPR motif's 6′ and 1′ amino acid positions are involved in RNA recognition [58]. In the presence of target RNA, structural studies revealed that the PPR repeats in PPR protein ZmPPR10 have a right-handed helical structure in the free state but undergo significant structural changes to become an anti-parallel homodimer molecule (Figure 3) [59]. With a few additional non-PPR protein factors, such as multiple organellar

RNA-editing factors (MORF) and organelle RRM proteins (ORRMs) proteins, the PPR–RNA complex is organized into a large editing complex known as an editosome. It is still unclear exactly how many protein factors are involved in the editosome's formation as well as their nature [28]. Additionally, it has been hypothesized that locating undetected RNA editing sites in the plant genome can be accomplished by comprehending the code that these PPR proteins use to recognize their targets [23]. As controllers of productivity modulation, links between site-specific PPR proteins and other proteins, or an undiscovered editing protein, these non-PPR proteins may be essential to editing [28].

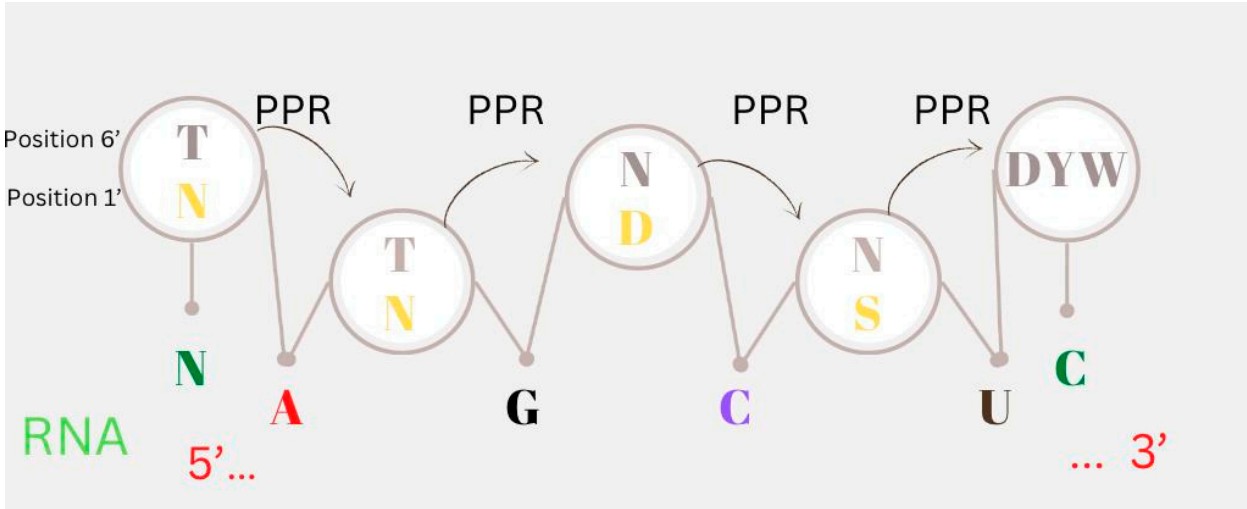

**Figure 3.** RNA editing and method of recognition code in RNA binding. Each PPR motif's amino acid position 6′ and 1′ in PLS repeats is denoted by a circle. To identify the recognition site, the binding to particular mRNA bases (upstream of the edited nucleotide) is specified by amino acid combinations at positions 6′ and 1′ of PPR (T, N) (T at 6′, N at 1′) binding to guanine (G), adenine (A), cytidine (C), cytidine (D), uridine (U), and (N, N) to either C or U. Tyrosine, asparagine, aspartic acid, and serine are each represented by the letters T, N, D, and S, respectively. The DYW domain has deaminase activity to change cytosine to uracile.

Apparently, around 200 PLS-type PPR proteins found in *Arabidopsis thaliana* can be attributed to the process of RNA editing [11,60]. Given the huge (600 in *Arabidopsis thaliana*) number of RNA edits, one would assume that multiple RNA editing sites must be edited by the editing PPR proteins [11]. The mitochondria, chloroplasts, or both are the primary targets of PPR proteins and non-PPR editing components [60]. This might indicate that the vital machinery for C-to-U editing might be preserved among both organelles. This editosome model is based on evidence mostly from data in *Arabidopsis thaliana*, but it might be possible that RNA editing in other plants might happen without the involvement of complex machinery (such as the editosome) and a single PPR-DYW editing protein with a few other unidentified non-PPR altering components might be enough [11].

Studies on loss-of-function mutants for specific PPR proteins suggest that the loss of one PPR protein does not completely inhibit RNA editing but rather attenuates it. This suggests cooperativity between PPR proteins inside a plant for the RNA editing process [42] For example, DYW-type PPR proteins such as RARE1 and VAC1 target the same site (accD gene) for RNA editing in Arabidopsis thaliana. Gene disruption for RARE1 led to no editing in the target, while the same for VAC1 resulted in a 60% reduction, suggesting its role might not be indispensable for RNA editing in the accD gene [61]. Some studies also showed the binding of DYW domains to zinc ions for Arabidopsis thaliana and in mosses.

## 6. Bioinformatics Approaches to Studying RNA Editing

Rapid and advanced genome sequence techniques have provided access to high-resolution and high-throughput data, which has equipped researchers to obtain more insights into RNA editing events [62]. Given the size and complexity of the data available, it is prudent to use bioinformatics and big data analysis (Figure 4) to derive useful information about RNA editing in plants [15]. These attempts involve the generation and curation of databases, the use of prediction tools, big data analysis for high-throughput sequencing data, artificial intelligence, and support vector machine or machine learning methods as learning resources [18].

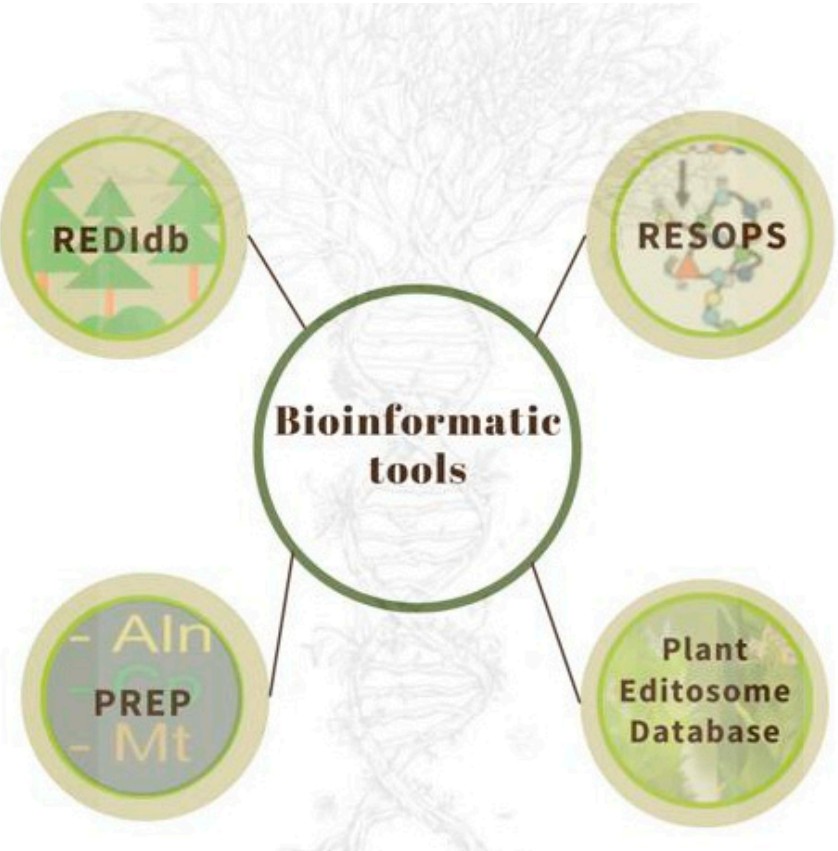

**Figure 4.** Few bioinformatic tools useful in data analysis for conducting research on RNA editing in plants.

The first resource is the databases where information about RNA editing in plants has been curated and maintained in a user-friendly format for usage by the researchers. Such resources include a general-purpose database that categorizes based on GO categories and genome annotation databases, but we mention databases that specifically focus on RNA editing events and associated information. One such database is the RNA Editing Site on Protein 3D Structures (RESOPS) [63], which contains unedited and edited versions of RNA editing targets and 3-D protein structure displays for RNA editing sites to assist researchers in understanding the role of editing in plants. Another database is the RNA Editing Database in Plants (REDIdb) [64], which is a comprehensive collection of more than 26,000 RNA editing entries that are manually curated with multiple information about RNA editing and the related organism to help researchers study the RNA editing events in an evolutionary context. Another useful database is the Plant Editosome Database (PED) [65], where plants' RNA editosome and associated data are curated and made available for search and information exchange.

The second resource is prediction tools, which try to narrow down the search for RNA editing sites and provide top hits for these sites. The Predictive RNA Editors for Plants (PREP) suite [65] is one such tool that uses sequence alignment with homologous proteins from other plants to predict C-U RNA edit sites in embryophyte mitochondrial and chloroplast transcripts. C-to-U RNA Editing in Chloroplasts (CURE-Chloroplast) [66] is another prediction tool that can process the entire chloroplast genome or just individual genes as a query to find C-to-U editing sites in seeded plant chloroplasts. The Plant RNA Editing Prediction and Analysis Computer Tool (PREPACT) [67] is another tool that allows finding RNA editing sites and helps to compare phylogenies by using protein sequences from primary or secondary protein databases or cDNA sequences from the plant genome.

The third resource is the use of data analysis methods to analyze high-throughput sequencing (HTS) data by comparing the cDNA sequences with their respective genomic loci. Many such tools exist, but only two tools, 'REDItools' and 'ChloroSeq', which are specific to C-to-U and U-to-C edits in plant organelles, have been developed. REDItools uses RNA-Seq and DNA-Seq data as input from the same sample and identifies editing sites by performing a genome-wide search for RNA editing [68]. On the other hand, ChloroSeq can perform efficient splicing, gene expression analyses, and RNA editing profiling by using RNA-Seq data as input [69]. Another tool is the RNA Editing Site Detector (RED), which detects RNA editing sites at the genome level with a filtered list of pre-calculated RNA variants and provides a graphical user interface output for easy visualization and comparison [70]. Another such tool is RNA Editing Detection in Organelles (REDO) [71], which identifies RNA editing sites using reference genome sequences, transcriptome data, and gene annotations. RNA Editing Site Scanner (RES-Scanner) [72] identifies RNA editing sites using RNA-seq and whole-genome sequencing data.

The fourth resource involves machine learning (ML) approaches, where statistical methods progressively improve the performance of computer systems to predict or analyze RNA editing events [67]. This is a conglomerate of interdisciplinary efforts spanning statistics, biology, and bioinformatics and is a constantly developing field that relies heavily on databases. Thus, the ML method being used is as good as the available information in the database (training datasets) and the level of accuracy in the underlying algorithm. Many statistical approaches are used to predict C-to-U RNA editing sites, but the main ones use a tree-based statistical model and random forest analysis [73,74]. Another approach called REGAL uses a heuristic search strategy to look for the most 'fit' hits based on the training of a dataset of known editing sites [75]. A similar method called C-to-U RNA Editor (CURE-Chloroplast) is based on a support-vector machine algorithm where C-to-U edit sites are identified based on similarity to previous datasets (based on data from *B. napus*, *O. sativa* and *A. thaliana*), and a weighted scoring model is used to provide scores for the top hits [67].

Finally, many future explanations of some events should be clarified, such as how prymidine changes to purines, such as G to C or U to G. Additionally, we should investigate how we can exploit this phenomenon to improve the tolerance of plants to environmental stress after discovering the relationship of RNA editing with endurance to adverse environmental conditions [2,12,13].

## 7. Conclusions

The transfer of genetic data from DNA to RNA and proteins forms the central dogma in biology, but certain deviations exist, and RNA editing is one of the prominent causes. RNA editing has many mechanisms, but nucleotide substitution is the most common in plants and has been found extensively in mRNA, while it is less common in tRNA and rRNA. Although the most common type of RNA editing in flowering plants is C to U, other types have been found, such as U to C or U to G. RNA editing in plants is very important and is evident due to rampant editing events inside mitochondria and chloroplast genes, which are vital for plant embryogenesis, pigment development, growth and coping with stress. Recent advancements in our knowledge of the involved proteins, such as PPR

proteins, high-throughput nucleic acid sequencing, and bioinformatics tools to predict and analyze RNA editing events have provided key insights into the importance of RNA editing in plant homeostasis and stress response. The insights derived from studying RNA editing have huge applications in policy design and mitigation strategies useful in scientific research and have translational value for agriculture.

**Author Contributions:** Conceptualization, T.M.; investigation and resources, A.F.; data curation and writing—original draft preparation, A.M.R. All authors have read and agreed to the published version of the manuscript.

**Funding:** This research received no external funding.

**Institutional Review Board Statement:** Not applicable.

**Informed Consent Statement:** Not applicable.

**Data Availability Statement:** Not applicable.

**Conflicts of Interest:** The authors declare no conflict of interest.

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
