# Peer review of "RNA Editing in Chloroplast: Advancements and Opportunities"

_cimb, doi:10.3390/cimb44110379_

Round 1

Reviewer 1 Report

Dear Editor, 

Thank you for providing me with an opportunity to review article for your prestigious Journal. The article is interesting and provide very detail information on title discussed. I have few minor comments for the authors to incorporate before acceptance. 

1. Be consistent with the abbreviation, the full form must appear first, later abbreviation can be used. 

2.  Some key references are missing, large statements need references. 

3.  The objectives of the study are not clear, revise them for general understanding. 

4. The future direction is missing which need to be incorporated in the revised version.

Author Response

Reviewer 1

  1. Be consistent with the abbreviation, the full form must appear first, later abbreviation can be used. 

Response

Thankyou, Done

  1. Some key references are missing, large statements need references.

Response

Thankyou, Done 

  1. The objectives of the study are not clear, revise them for general understanding. 

Response

Thankyou, I rephrase the end of introduction to clarify objectives

  1. The future direction is missing which need to be incorporated in the revised version.

Response

Thankyou, I add statement before conclusion stated the future direction

Reviewer 2 Report

The review article entitled "RNA Editing In Chloroplast: Advancements and Opportunities" is an interesting topic and may attract broad readership from plant scientists working in different domains. 

However, there are many concerns within the review which does not permit its acceptance in the current form. Some blunder scientific errors include not mentioning the scientific names of plants in italics. Figure legends does not tell clearly about the facts mentioned in figure. For example in Fig. 1, there is no mention in figure legends about the factors or promoter locations mentioned in the figure.  Why are the NEPs have different types: type 1a and type 1b.  Figure 1b needs to be redrawn and does not tell anything about plant transcription. There are different kinds of colored rectangles and hexagons drawn within the figures along with colored circular ribosomes. None of this have been labelled in the figure.  There is no mention where the choloroplast is within the figure. 

Other major issues within the entire review: 

Abstract: The abstract should start with a line about RNA editing. It just state that it is an important event in plants but nothing in general. 

Introduction: Technical jargons like editosomes are used without any details. Instead of drawing general plant transcription, there needs to be a general figure on RNA editing mechanism.

Line 39-40: There is only one reference although the line states that numerous studies have been done. 

Line 60: NEP in Fig.1a is nuclear encoded but in Line 60 it is written as phage like RNA Polymerase. Also, in what way it is similar to phage polymerase. 

Figure legends does not relate to the figures. 

Line 97/Line 132/133 and many other places scientific names are not written in italics. 

Line 98 is faulty. It should be analysis of .

Line 125 needs to be rephrased. 

Figure 2 legends does not explain anything about the figure motifs. 

Mechanism of RNA editing is highly confusing and the different concepts abruptly.  Figure 3 is unclear again and needs to be redrawn.

Figure 4 just contains icons from different software. This seems more like plagiarism. The authors of the review article should redraw these and input some thoughts on their part to make it creative. 

Conclusions talks on a very limited level about the opportunities. 

Author Response

First of all,   I would like appreciate the respective reviewer to help us to improve our manuscript

The review article entitled "RNA Editing In Chloroplast: Advancements and Opportunities" is an interesting topic and may attract broad readership from plant scientists working in different domains.

  • However, there are many concerns within the review which does not permit its acceptance in the current form.

Some blunder scientific errors include not mentioning the scientific names of plants in italics.

Response

Thankyou, Done

Figure legends does not tell clearly about the facts mentioned in figure. For example in Fig. 1, there is no mention in figure legends about the factors or promoter locations mentioned in the figure.

Response

Thankyou, we add some details information to thefigureʼs legend and stated it in text

 Why the NEPs are have different types: type 1a and type 1b.

Response

These depend on recognized promoter formation. Most NEP promoters have a core sequence motif (YRTA; type-Ia), similar to promoters of plant mitochondria. A subclass of NEP promoters shares a GAA-box motif upstream of the YRTA-motif [type-Ib]. Type-II NEP promoters, represented by dicot clpP promoters, lack these motifs and possess crucial sequences located downstream of the transcription initiation site. I stated the reference in the text

 Figure 1b needs to be redrawn and does not tell anything about plant transcription. There are different kinds of colored rectangles and hexagons drawn within the figures along with colored circular ribosomes. None of this have been labelled in the figure.  There is no mention where the choloroplast is within the figure.

Response

Thankyou. I changed the figure to clarify what I mean in chloroplast transcription

Other major issues within the entire review:

Abstract: The abstract should start with a line about RNA editing. It just state that it is an important event in plants but nothing in general.

Response

Thankyou so much, Done

Introduction: Technical jargons like editosomes are used without any details. Instead of drawing general plant transcription, there needs to be a general figure on RNA editing mechanism.

Response

Thankyou, I add two paragraphs to clarify RNA editing mechanisms

Line 39-40: There is only one reference although the line states that numerous studies have been done.

Response

Thankyou, I add two refrences more

Line 60: NEP in Fig.1a is nuclear encoded but in Line 60 it is written as phage like RNA polymerase. Also, in what way it is similar to phage polymerase.

Response

Yes, although it is encoding from the nucleus genome but it is like phage as a monomeric polymerase, in other words it is not consist of subunits   

Figure legends does not relate to the figures.

Response

Thankyou, it is improved

Line 97/Line 132/133 and many other places scientific names are not written in italics.

Response

Thankyou, Done  

Line 98 is faulty. It should be analysis of .

Response

Thankyou, Done

Line 125 needs to be rephrased.

Response

Done

Figure 2 legends does not explain anything about the figure motifs.

Response

Thankyou, more details are added

Mechanism of RNA editing is highly confusing and the different concepts abruptly.  Figure 3 is unclear again and needs to be redrawn.

Response

Thankyou, the figure is redrawn

Figure 4 just contains icons from different software. This seems more like plagiarism. The authors of the review article should redraw these and input some thoughts on their part to make it creative.

Response

Thankyou, the figure is redrawn

Conclusions talks on a very limited level about the opportunities.

Thankyou, I tried to add some information to improve it
